# The mitochondrial DNA polymerase gamma degrades linear DNA fragments precluding the formation of deletions

Nadee Nissanka [1], Sandra R. Bacman[2], Melanie J. Plastini[1] & Carlos T. Moraes[1,2]

Double-strand breaks in the mitochondrial DNA (mtDNA) result in the formation of linear fragments that are rapidly degraded. However, the identity of the nuclease(s) performing this function is not known. We found that the exonuclease function of the mtDNA polymerase gamma (POLG) is required for this rapid degradation of mtDNA fragments. POLG is recruited to linearized DNA fragments in an origin of replication-independent manner. Moreover, in the absence of POLG exonuclease activity, the prolonged existence of mtDNA linear fragments leads to increased levels of mtDNA deletions, which have been previously identified in the mutator mouse, patients with *POLG* mutations and normal aging.

[1] Neuroscience Graduate Program, University of Miami Miller School of Medicine, Miami, FL 33136, USA. [2] Department of Neurology, University of Miami Miller School of Medicine, Miami, FL 33136, USA. Correspondence and requests for materials should be addressed to C.T.M. (email: cmoraes@med.miami.edu)

The mammalian mitochondrial DNA (mtDNA) codes for critical subunits of the mitochondrial oxidative phosphorylation system. Large deletions in the mtDNA have been associated with human diseases and normal aging[1]. The molecular mechanisms leading to the formation of mtDNA deletions are not well understood, but defective replication[2,3], repair[4], replication stalling[5], and double-strand breaks (DSBs)[6] have been suggested to mediate the formation of large rearrangements, particularly deletions. The impact of DSBs in forming mtDNA deletions is minimized by the fact that linearized mtDNA fragments have a very short half-life in cells[7], being rapidly degraded by unknown nucleases[8]. This rapid elimination has facilitated the use of mtDNA editing enzyme-mediated cleavage of mutant mtDNA to shift heteroplasmy, as the undigested mtDNA genomes in heteroplasmic cells can repopulate the organelles after DSBs and eliminate the mutant mtDNA[9,10].

The nature of these putative nucleases has been explored recently, but none of the tested mitochondrial nucleases appeared to be effective in eliminating mtDNA fragments after DSBs[8]. Curiously, the mtDNA mutator mice, which harbor an exonuclease/proofreading inactive catalytic subunit of the mtDNA polymerase gamma ($Polg^{D257A/D257A}$), showed the presence of an 11 kb linear mtDNA fragment which corresponds to most of the mtDNA major arc (the longer DNA region between origins of heavy ($O_H$) and light ($O_L$) strand replication)[11]. These mice also have an increased rate of mtDNA point mutation formation and, more importantly, an accumulation of multiple large deletions which were suggested to be the driving force behind their premature aging phenotype[11–13].

In addition, mutations in *Polg* have been associated with multiple mtDNA deletions in humans[14]. However, mutations in other mtDNA replication factors also lead to the formation of circular mtDNA deletions (reviewed in ref. [15]). We now found that POLG minimizes the formation of mtDNA deletions by rapidly eliminating linear fragments.

## Results

**Expressing mitochondrial-targeted endonucleases**. We have used recombinant adenoviruses (Ad) expressing two mitochondrial-targeted restriction enzymes, mito*Sca*I-HA and mito*Apa*LI-HA (Fig. 1a)[16,17]. These adenoviruses are effective in promoting DSBs in mtDNA in mice heteroplasmic for BALB and NZB haplotypes[16,17]. We used these adenoviruses to express mitochondrial restriction endonucleases in fibroblast lines derived from lung of the mutator and wild-type mice. Fibroblasts were infected with the respective adenovirus and samples collected at 1, 2, 5, and 10 days after infection. Because these mitochondrial-targeted endonucleases contain an HA tag, we could confirm the mitochondrial localization of mito*Sca*I-HA and mito*Apa*LI-HA, which as expected, were strongly expressed in mitochondria after 1-2 days but decreased by day 10 (Fig. 1b, Supplementary Fig. 1).

In addition, we also injected recombinant adenovirus retro-orbitally in the mutator and control mice. The retro-orbital sinus drains directly into the venous system, providing an effective alternative to tail vein injections[18]. Adenovirus has a strong tropism for hepatocytes, and recombinant adenoviruses are highly expressed in liver[19]. Accordingly, we found that Ad-*GFP* was strongly expressed in liver 5 days after retro-orbital injections (Fig. 1c). We next injected recombinant adenovirus expressing mito*Sca*I-HA or mito*Apa*LI-HA in the retro-orbital sinus of 20-day-old mice. Controls were injected with Ad-*GFP*. Livers analyzed 5 days after injection showed strong mitochondrial expression in hepatocytes (Fig. 1d).

**Delayed degradation of mtDNA fragments in the mutator mouse**. In the C57BL6/J mouse mtDNA there are 3 *Sca*I sites, whereas *Apa*LI has a single site (Fig. 2a). We first analyzed DNA from fibroblasts infected with Ad-mito*Sca*I-HA and Ad-mito*Apa*LI-HA. Southern blot analyses showed that the uninfected and undigested DNA from the mutator mouse naturally contained mtDNA fragments that were smaller than the 16.3 kb wild-type (Fig. 2b). This feature had been already reported in the mutator mouse[11,20]. Fibroblasts were infected with Ad-mitoRestriction Endonucleases and samples were analyzed at 1, 2, 5, and 10 days after infection. To better visualize the mtDNA fragments in Southern blots, after purification aliquots of total DNA were digested with *Sac*I, which linearizes the mtDNA at position 9045 (Fig. 2a). As previously observed, fragmented DNA is rapidly degraded and cannot be easily detected in wild-type fibroblasts[7]. Instead, a severe depletion of the full-length 16.3 kb genome was observed (Fig. 2b). In contrast, fibroblasts of the mutator mouse accumulated higher levels of the expected linear fragments at early time points (Fig. 2b). These fragments were ultimately eliminated at later time points.

Infection of fibroblasts with Ad-mito*Apa*LI-HA produced a single DSB, but the results were essentially identical to the ones obtained with the 3-site mito*Sca*I (Fig. 2c). Again, wild-type fibroblasts showed a strong depletion followed by repopulation with full-length mtDNA at later time points, when the levels of expression of the mito*Apa*LI-HA decreased (Figs. 1a and 2c). In contrast, the mutator fibroblasts showed markedly delayed degradation of the linear mtDNA fragments. Previously existing endogenous fragments (likely modified by mito*Apa*LI) were clearly visualized in the mito*Apa*LI-HA blots (marked with an * in Fig. 2). Similar results were obtained with fibroblasts infected with a lentivirus expressing mito*Pst*I (Supplementary Fig. 2). This confirms that fragments lacking either or both $O_H$ or $O_L$ origins of DNA replication are eliminated at comparable rates.

To ensure that the observations with cultured fibroblasts also occur in vivo, we analyzed mtDNA from the liver of mice systemically injected with the recombinant adenovirus. DNA was purified and post-digested with *Sac*I for linearization (Fig. 2d). Wild-type control mice, which were injected with Ad-*GFP*, showed a single full-length linearized band. As expected, liver mtDNA of the mutator mouse, even without injection, had natural smaller mtDNA species, which likely corresponds to previously described linearized molecules[11,20]. MtDNA from liver samples extracted from mice 5 days after injection of Ad-mito*Sca*I-HA or Ad-mito*Apa*LI-HA showed a band pattern similar to what was observed with fibroblasts in culture. Five days after injection we could still clearly detect the predicted linear fragments resulting from the DSB caused by the mitochondrial restriction endonucleases (Fig. 2d).

The results obtained by Southern blot were confirmed by quantitative PCR (qPCR). Quantitation of the ratios of a mtDNA gene (ND1) to a nuclear gene (β-Actin) showed a rapid and marked mtDNA depletion in the wild-type fibroblasts and liver samples using either Ad-mito*Sca*I-HA or Ad-mito*Apa*LI-HA infections (Fig. 3a, b). On the other hand, samples from the mutator mouse showed more persistent levels of linearized fragments containing the ND1 gene (Fig. 3c, d). As expected, our qPCR data suggests that the smaller ND1-containing fragment in in the Ad-mito*Sca*I-HA infected wild-type fibroblasts underwent a more rapid depletion than the larger corresponding fragment in the Ad-mito*Apa*LI-HA infected wild-type fibroblasts. At earlier time points after Ad-mito*Sca*I-HA infection, the smaller D-loop-containing fragment underwent a faster depletion than the larger ND1-containing fragment in both the wild-type and mutator fibroblasts (Fig. 3e).

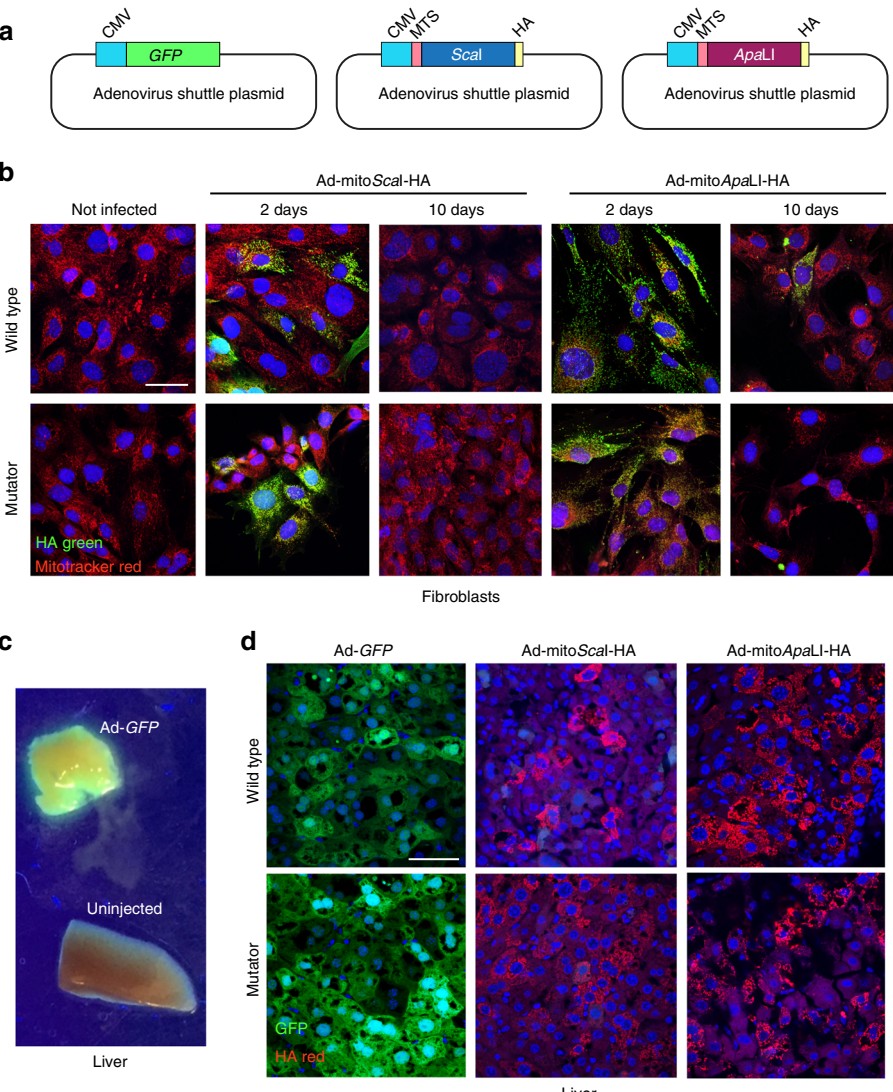

**Fig. 1** Expression of mitochondrial-targeted restriction endonucleases in the mutator mouse. **a** Diagrams illustrating the basic gene components of the plasmids used to make adenoviral particles. **b** Lung fibroblasts from wild-type and mutator mice were established as described in methods. Fibroblasts were infected with adenovirus expressing mitoScaI-HA or mitoApaLI-HA restriction endonucleases. Fibroblasts were stained with Mitotracker Red and fixed after 1, 2, 5, and 10 days. See Supplementary Fig. 1 for days 1 and 5. Fibroblasts were immuno-stained for HA. Scale bar = 50 μm. **c** C57BL6/J mice were injected with adenovirus expressing GFP. After 5 days mice were killed and livers exposed to UV light. **d** Mutator and wild-type mice were injected with adenovirus expressing GFP, mitoScaI-HA or mitoApaLI-HA. After 5 days mice were killed and the livers removed and subjected to immunostaining for HA. Scale bar = 50 μm

To confirm that the exonuclease activity does not depend on the polymerase activity of POLG for the delayed degradation of the fragmented mtDNA following DSBs, we independently cloned the wild-type mouse POLGA subunit and a mutant version harboring the R920H missense mutation in the polymerase domain, into lentiviral vectors. The mouse POLGA R920H mutation is the homolog of the previously described human POLGA R943H mutation found in patients with mtDNA depletion[21]. Wild-type and mutator fibroblasts were infected with each lentivirus, and 5 days after lentiviral expression, cells were infected with Ad-mitoScaI-HA and collected at 1, 2, 5, and 10 days post-infection. Quantification of the relative levels of mtDNA showed similar levels of depletion following DSBs in the lenti-WT-*Polg* and lenti-R920H-*Polg* infected cells, suggesting that the polymerase activity of POLG is not required for the fast degradation of the linear mtDNA fragments (Supplementary Fig. 3).

**Delayed mtDNA degradation increases deletion levels**. To test whether the increased availability of free DNA ends affects the formation of mtDNA rearrangements, we tested for the presence of deletion breakpoints in our samples. Based on our previous experience, free mtDNA ends tend to recombine, albeit at very low levels[17,22]. Therefore, we took advantage of the fact that ScaI has distant recognition sequences in the mtDNA that could ligate and be detected by amplifying the deletion breakpoint surrounding the cleavage sites. We attempted to amplify three putative breakpoints surrounding the ScaI sites. The oligonucleotides used in the amplification are indicated in Fig. 4a, b as colored arrows. One of the amplifications, would originate from recombinant molecules lacking the origin of heavy-strand replication. These [deletions without $O_H$] would not be expected to replicate and would be short lived. In agreement, we found an increase in the levels of this amplicon 1, 2, and 5 days after Ad-mitoScaI-HA expression in fibroblasts, followed by a decrease by

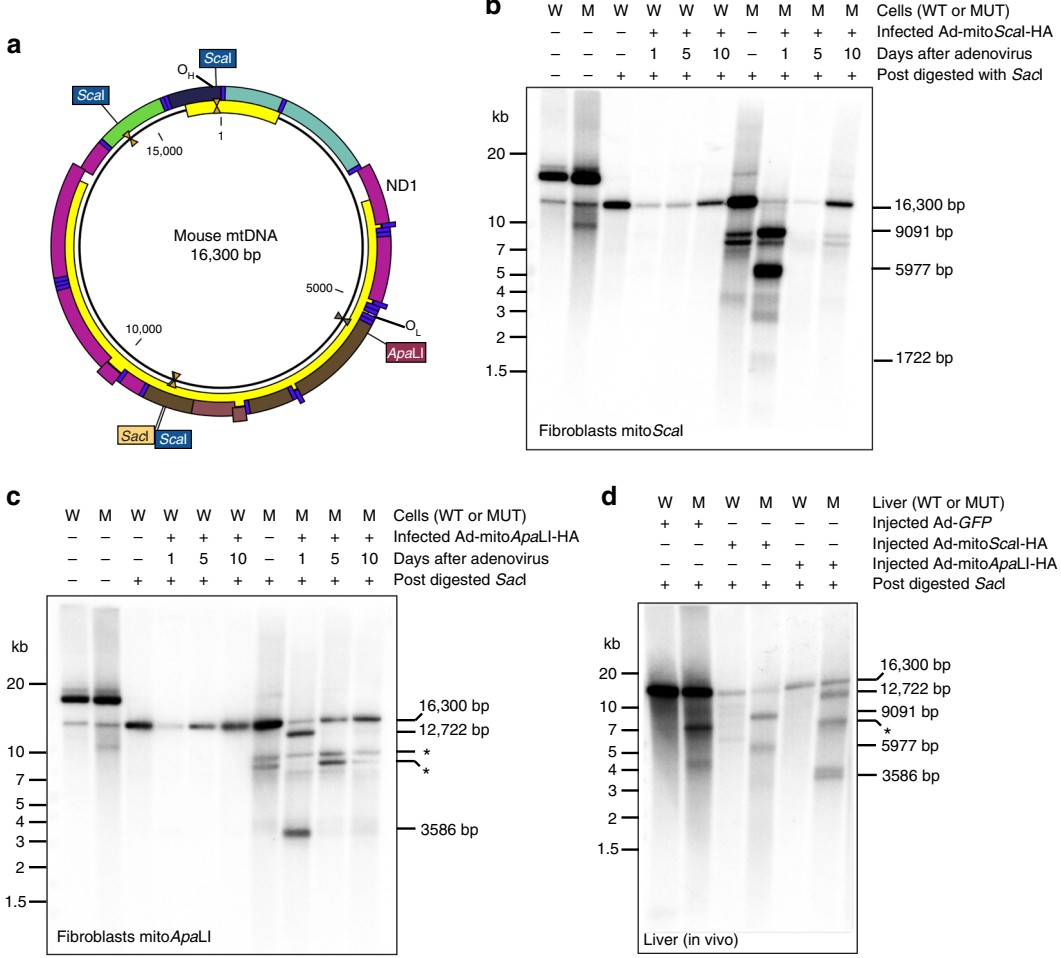

**Fig. 2** Delayed degradation of mtDNA after DSB in DNA from fibroblasts and the liver of the mutator mouse. **a** Diagram of wild-type C57BL6/J mtDNA illustrating the location and number of restriction endonuclease sites. Southern blot from mutator and wild-type fibroblasts infected with (**b**) Ad-mito*Sca*I-HA and (**c**) Ad-mito*Apa*LI-HA at 1, 5, and 10 days post-infection. **d** Southern blot from mutator and wild-type mice liver 5 days after retro-orbital injections with Ad-*GFP*, Ad-mito*Sca*I-HA, or Ad-mito*Apa*LI-HA. DNA was post-digested with *Sac*I to linearize mtDNA. Zeta-Probe membrane was hybridized with a [α-$^{32}$P] dCTP labeled probe covering 11.7 kb of mtDNA

day 10 (Fig. 4c). There were essentially no amplifications from DNA purified from wild-type fibroblasts (Fig. 4c). The second amplicon would originate from rearranged mtDNA molecules retaining $O_H$ but lacking $O_L$. In this case, we also observed a greater increase in the formation of rearrangements in the mutator samples, particularly at earlier time points. Again, the levels of these rearranged molecules decreased with time. (Fig. 4d). The third amplicon was from a recombined molecule originated from a partially digested mtDNA. In this case, where both $O_H$ and $O_L$ were present, the levels of the mutant molecule continue to accumulate with time (Fig. 4e). In liver, we detected a clear increase of all types of *Sca*I-mediated rearrangements in the mtDNA at the only time point analyzed, 5 days after injection (Fig. 4f–h).

Sequence analyses of the rearrangement breakpoints after mito*Sca*I exposure showed that breakpoints of $O_H$- and $O_H/O_L$-containing deleted molecules in the mutator mtDNA were not qualitatively different from the ones observed in the mtDNA from wild-type mice (Fig. 5). In both cases, the breakpoints were close to but not immediately adjacent to the *Sca*I sites. Some had micro or imperfect homologies in the breakpoint regions, whereas others did not (Fig. 5, Supplementary Figs. 4 and 5). Accordingly, the distance of the breakpoint from the *Sca*I site (chew-back), was

not different between deletions from wild-type and mutator fibroblasts or liver.

## Discussion

For more than 15 years we have known that mtDNA is rapidly eliminated after a DSB[6,7,23]. This feature has a number of potential critical physiological roles, including: (1) assure that mtDNA binding proteins associate only with viable circular molecules[24]; (2) avoid the release of mtDNA from cells, which are known to function as damage-associated molecular patterns (DAMPs), triggering inflammation[25,26]; and (3) Avoid the presence of DNA free ends that can potentially lead to rearrangements[22]. Because a single cell contains approximately 1,000 copies of mtDNA, cells have developed mechanisms to eliminate damaged and linearized fragments. However, these mechanisms were unknown. A recent study surveyed most known mitochondrial nucleases, but could not find one that was responsible for the elimination of mtDNA after DSB[8].

The present study showed that the exonuclease activity of POLG has an important role in eliminating fragmented mtDNA from mitochondria. Worth noting, wild-type POLG eliminated fragments lacking the origin of heavy-strand replication ($O_H$), the

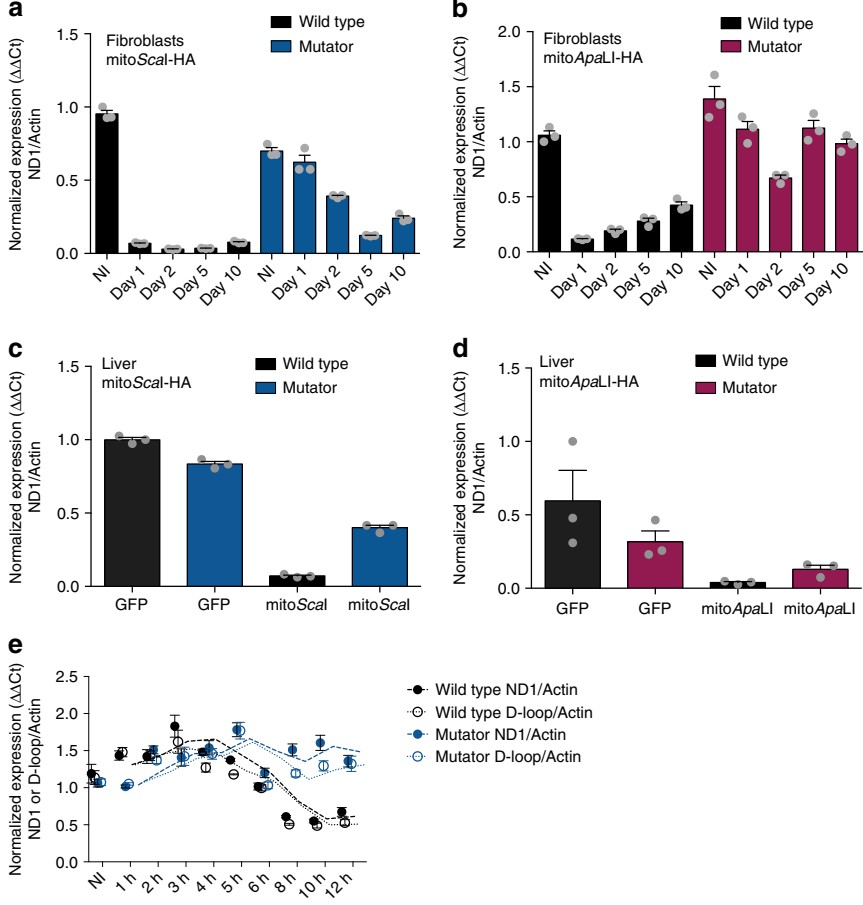

**Fig. 3** Quantification of mtDNA levels after DSB in the mutator mouse. Relative levels of total mtDNA were assessed using ND1 (mtDNA) and β-Actin (nuclear DNA) primers in **a** mutator and wild-type fibroblasts infected with Ad-mito*Sca*I-HA at 1, 2, 5, and 10 days post-infection; **b** mutator and wild-type fibroblasts infected with Ad-mito*Apa*LI-HA at 1, 2, 5, and 10 days post-infection; **c** mutator and wild-type liver of mice infected by retro-orbital injection with Ad-*GFP* or Ad-mito*Sca*I-HA at 5 days post-injection; **d** mutator and wild-type liver of mice infected by retro-orbital injection with Ad-mito*Apa*LI-HA at 5 days post-injection. **e** Relative levels of total mtDNA were assessed using ND1 or D-loop (mtDNA) and β-Actin (nuclear DNA) primers in mutator and wild-type fibroblasts infected with Ad-mito*Sca*I-HA at 1, 2, 3, 4, 5, 6, 8, 10, and 12 h post-infection. Dashed line indicates the moving average for the two time points flanking the time point of interest. NI indicates not infected. Quantification was performed using comparative ΔΔCt method. Error bars (s. e.m.) derive from technical replicates

one widely accepted recruiting/docking site for POLG[27–29]. If the POLG molecules performing DNA degradation are not engaged in replication, the POLG degradative function would be independent from its replicative one. In addition, we have also shown that the polymerase activity of POLG does not play a role in degrading linear mtDNA fragments. Therefore, origins of replications do not appear to be required for POLG to dock at DNA ends as an exonuclease. If this non-specific docking occurs during replication, it would provide support for the coupled mtDNA replication model[30]. However, this POLG/DNA docking may occur only at free double-stranded ends for 3′-5′ degradation and not at replication competent circles for DNA polymerization.

POLG appears to have evolved to perform different functions. It was shown that the exonuclease activity is also required for efficient DNA ligation after replication, as well-suited flaps are not formed in the mutator mouse[20]. POLG was also found to be required for elimination of paternal mtDNA during spermatogenesis in Drosophila[31]. Surprisingly, the exonuclease activity was not required for this function, suggesting that the mechanism may not be the same as the one we describe here for mammalian somatic cells. POLG was recently reported to degrade mtDNA in yeast as a response to autophagy signals[32]. These reports illustrate a remarkable functional flexibility for POLG.

Clearly, POLG is not the only nuclease in charge of eliminating fragmented mtDNA, as these fragments eventually disappeared, at later time points. Other factors, involved or not in replication of mtDNA may also be involved in the degradation of linear fragments. Recently, MGME1 was found to also degrade mtDNA fragments and in its absence, increased mtDNA deletions were detected[33].

The existence of other factors make it difficult to explain the presence of linear mtDNA in tissues of the mutator mouse[20,34], suggesting that these different players may work together. It is unclear why these alternative nucleases would not eliminate linear DNA fragments. Potentially, they could be eliminated, but because they are formed frequently, their steady-state levels are maintained. Macao and colleagues suggested that the impairment of DNA ligation by exonuclease-deficient POLG could lead to nicks, DSB and formation of linear fragments during mtDNA replication[20]. At the moment, we do not have answers to many of these questions.

Besides linear fragments, tissues of the mutator mouse also accumulate circular mtDNA deletions with age[11,13]. In fact, it has been proposed that the accumulation of mtDNA deletions is the best predictor for the accelerated aging phenotype[12]. The prolonged presence of free DNA ends is a known contributor for

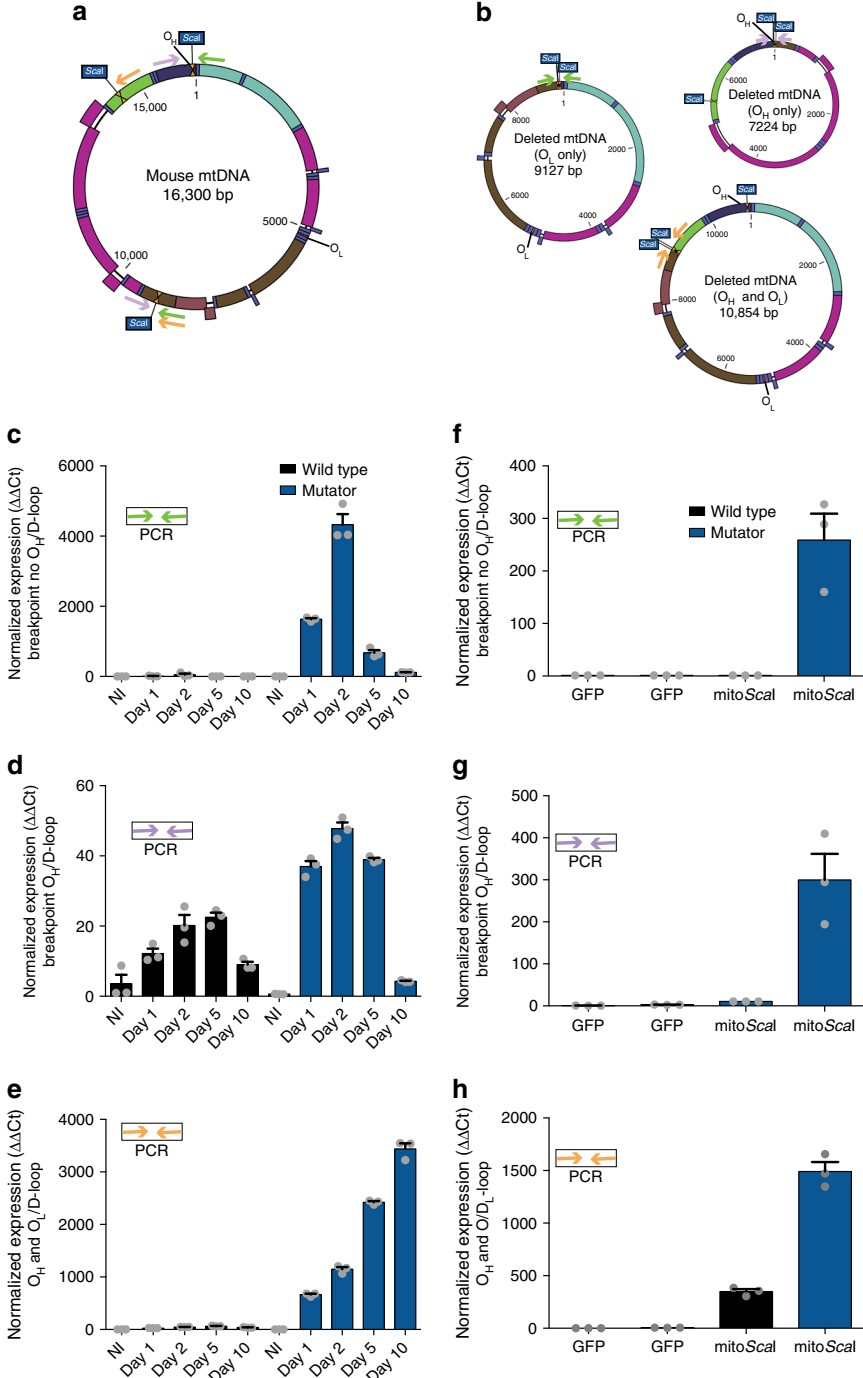

**Fig. 4** Quantification of mtDNA rearrangements after DSB in the mutator mouse. **a** Diagram of wild-type C57BL6/J mtDNA illustrating the location and number of *Sca*I sites and primers used to quantify rearrangements following *Sca*I infection. Green primers amplify rearrangements that do not contain the origin of replication of the heavy-strand (O_H), purple primers amplify rearrangements that contain the O_H, orange primers amplify rearrangements that contain the O_H and the O_L. **b** Diagrams of putative C57BL6/J partially-deleted mtDNA after Ad-mito*Sca*I-HA infection and primers used to quantify rearrangements. One deleted molecule does not contain the O_H (green primers) whereas the other deleted molecule contains the O_H (purple primers). The last deleted molecule contains both the O_H and the O_L (orange primers). Quantification of the breakpoint without O_H in (**c**) mutator and wild-type fibroblasts at 1, 2, 5, and 10 days and (**f**) mutator and wild-type liver at 5 days after Ad-mito*Sca*I infection. Levels of the breakpoint without O_H were normalized to a region in the D-loop. Quantification of the breakpoint with O_H in (**d**) mutator and wild-type fibroblasts at 1, 2, 5, and 10 days and (**g**) mutator and wild-type liver at 5 days after Ad-mito*Sca*I-HA infection. Levels of the breakpoint with O_H were normalized to a region in the D-loop. Quantification of the breakpoint with O_H and O_L in (**e**) mutator and wild-type fibroblasts at 1, 2, 5, and 10 days and (**h**) mutator and wild-type liver at 5 days after Ad-mito*Sca*I-HA infection. Levels of the breakpoint with O_H and O_L were normalized to a region in the D-loop. NI indicates not infected. Quantification was performed using comparative $\Delta\Delta$Ct method. Error bars (s.e.m.) derive from technical replicates

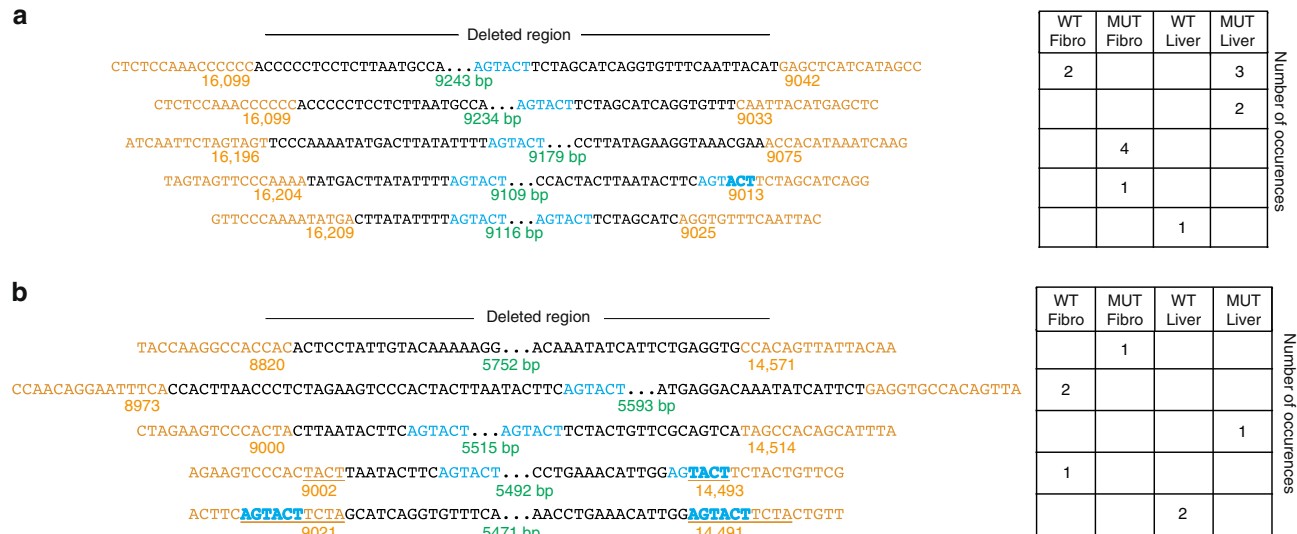

**Fig. 5** Sequence analyses of mtDNA deletion breakpoints. PCR amplifications of mtDNA deletion breakpoints from DNA of fibroblasts and liver infected with Ad-mito*Sca*I-HA (wild-type and mutator) were cloned in a TOPO TA plasmid (TA cloning) and sequenced using M13F and M13R primers. **a** Five representative sequences from the deletions [with O$_H$] and their frequency of detection in different samples. **b** Five representative sequences from the deletions [with O$_H$ and O$_L$] and their frequency of detection in different samples. Breakpoints are annotated as follows, orange text - mtDNA that is retained, black text -mtDNA which is deleted, blue text – *Sca*I sequence, green text - size of the deletion breakpoint, underlined text - micro or imperfect homologies

deletion formation by non-homologous end joining (NHEJ) in the nucleus[35]. There is no know DSB repair mechanism in mitochondria[36], but DSBs do promote the formation of large deletions in mtDNA[22]. In agreement, we found the presence of rearrangement breakpoints in fibroblasts and liver expressing mitochondrial-targeted restriction endonucleases. These levels were markedly higher in the mutator mice after DSB. As expected, we found that the increase in deletion breakpoint levels lacking either O$_H$ or O$_L$ went down at day 10, compared to days 1–2. On the other hand, deleted molecules containing the origins of replication continued to accumulate with time. Therefore, our results indicate that the extended periods of DNA free ends exposure increase the odds of mtDNA deletion formation, which can be propagated and accumulated during aging. The mechanism of re-circularization/recombination remains unknown. In agreement with our previous results, only small or imperfect repeats were present at some of the breakpoints, suggesting some form of NHEJ is involved in most of the recombination events. The lack of POLG exonuclease activity did not change the features of the deletion breakpoints, which included the loss of 0–200 bp from the free ends. If this loss is the consequence of exonuclease chew-back, we have to assume that it is performed by alternative exonucleases, as these losses were observed in both wild-type and mutator fibroblasts and liver.

Mutations in *POLG* have been extensively associated with mtDNA deletions in humans. However, these mutations are not restricted to the exonuclease domain and are mostly dominant[37,38]. This suggests that different mutations that disturb POLG conformation creates the conditions for the formation mtDNA deletions. Although we have no evidence, one could speculate that these mutations lead to conformational changes that may affect the independent exonuclease function, possibly by altering POLG interactions with specific partners.

Mutations in replication factors, including *TWNK* and *POLG*, both of which cause multiple mtDNA deletions, also increase the levels of age-related mtDNA deletions and it was suggested that it was associated with replication stalling, which can lead to DSBs[5]. Interestingly, mtDNA deletions have been found to be also increased in the substantia nigra of sporadic Parkinson's Disease patients[39]. Several patients with *POLG* mutations and multiple mtDNA deletions have presented with Parkinsonism[40–42]. In summary, the formation of mtDNA deletions in a vast array of clinical conditions may be associated with the impairment of mtDNA fragment elimination by POLG.

Another important aspect of mtDNA degradation after DSB is the recent development of therapeutic techniques to eliminate mutant mtDNA in heteroplasmic cells. Using DNA editing enzymes targeted to the mitochondria, mutant mtDNA is cleaved, whereas the wild-type is kept intact[9,10]. Because the wild-type genomes are commonly in the minority, it is critical that their levels increase to make up for the mutant elimination. This can only occur if the cleaved mtDNA is rapidly eliminated, so that the mtDNA copy number control can re-establish normal mtDNA levels using the residual wild-type genomes. Therefore, the importance of understanding the mechanisms of mtDNA elimination after DSB can help optimize this process.

## Methods

**Adenoviral preparation**. Mito*Sca*I–HA and mito*Apa*LI-HA were independently cloned into the pAdTrack5 adenoviral vector under the control of independent CMV promoters. Ad–mito*Sca*I–HA, Ad-mito*Apa*LI and control Ad–*GFP* adenovirus stocks were prepared by the Colorado State University Virus Core Facility (Ft. Collins, CO, USA). The adenovirus titers were estimated by OD260: Ad–mito*Sca*I–HA: 5.4 × 10$^{12}$ particles/mL; Ad–mito*Apa*LI–HA: 3.4 × 10$^{12}$ particles/mL; Ad–*GFP*: 9.2 × 10$^{12}$ particles/mL.

**Lentiviral preparation**. Wild-type *Polg* (wild-type exonuclease, wild-type polymerase domains) and R920H *Polg* (wild-type exonuclease, R920H polymerase domains) cDNAs were independently cloned into the pLenti-MP2 lentiviral vector under the control of CMV promoters. Lenti-WT-*Polg* and Lenti-R920H-*Polg* lentiviral stocks were prepared using LentiX Packaging Single Shots (Clontech).

**Animal procedures**. All mice procedures were performed according to a protocol approved by the University of Miami. Mice were housed in a virus/antigen-free facility at the University of Miami in a 12 hour light/dark cycle at room temperature and fed ad libitum with a standard rodent diet. The *Polg* mutator mice (from Jackson Laboratory: B6.129S7(Cg)-Polg$^{tm1Prol}$/J, https://www.jax.org/strain/017341) contains an alanine instead of an aspartate residue in the second exonuclease domain (D257A), resulting in an inactive exonuclease.

Retro-orbital injections were performed on anesthetized mice at 19–24 days. Mice were injected with $5 \times 10^{11}$ particles of either Ad–mito$Sca$I–HA, Ad–mito$Apa$LI–HA, or Ad–$GFP$ diluted to 40 µL in saline. Animals were killed 5 days after the retro-orbital injection. Anesthetized mice were transcardially perfused with ice-cold PBS. The liver was post-fixed overnight in 4% PFA, cryoprotected in 30% sucrose and frozen in OCT using isopentane. Another, unfixed liver fragment, was used for DNA purification and analyses.

**Cell culture procedures**. Primary cells from lung fibroblasts were generated in our laboratory from $Polg$ mutator mice and litter mate controls in high-glucose Dulbecco's modified Eagle medium supplemented with 10% fetal bovine serum, 1 mM pyruvate, and 50 µg/mL uridine at 37 °C in an atmosphere of 5% $CO_2$. Fibroblasts were immortalized by infection with a retrovirus expressing the E6 and E7 genes of type 16 Human Papilloma Virus carrying G418 resistance.

$Polg$ mutator and control fibroblasts were infected in culture with $3 \times 10^{11}$ particles of either Ad–mito$Sca$I–HA or Ad–mito$Apa$LI–HA. Viral particles were removed 1 day after infection. Fibroblasts were collected at 1, 2, 5, and 10 days post-infection for DNA extraction and immunocytochemistry.

$Polg$ mutator and control fibroblasts were infected in culture with either lenti-WT-$Polg$ or lenti-R920H-$Polg$. Viral particles were removed 1 day after infection. 5 days after lentiviral infection, fibroblasts were infected with Ad–mito$Sca$I–HA and collected at 1, 2, 5, and 10 days post-infection for DNA extraction.

**Southern blot**. Total DNA from liver and fibroblasts was extracted using phenol: chloroform/isopropanol precipitation. Two µg of total DNA was digested with $Sac$I (NEB), separated on a 0.75% agarose gel, and transferred to a Zeta-Probe membrane (Bio-Rad). Two templates:1.7 kb (covering the D-loop) and 10 kb (covering outside the D-loop) to detect mtDNA were amplified from genomic DNA from the cortex of a C57BL/6 J wild-type mouse. The following primers were used to amplify the mtDNA probes:

1.7 kb (F: CTATCCCCTTCCCCATTTGGTCTATTAAT; R: GTCATGAAATC TTCTGGGTGTAGG)

10 kb (F: GCCAGCCTGACCCATAGCCATAATAT; R: GAGAGATTTTATG GGTGTAATGCGG)

Amplified DNA was purified, labeled with [$\alpha$-$^{32}$P] dCTP using Random Primed DNA Labeling kit (Roche) and cleaned with G-50 Sephadex quick spin columns (GE Healthcare). Detection and densitometric quantification of mtDNA signals was performed with the Cyclone Plus Phosphor Imager equipped with the Optiquant software (PerkinElmer).

**Quantitative PCR**. Quantitative PCR reactions using SYBR chemistry (SsoAdvanced Universal Master Mix SYBR Green, Bio-Rad) or TaqMan chemistry (PrimeTime Std qPCR Assay, IDT) were performed on a Bio-Rad CFX96/C1000 qPCR machine. We used the comparative $\Delta\Delta$Ct method to determine the relative quantity of mtDNA[43]. The levels of different mtDNA species between non-infected and infected samples were determined by quantifying the levels of total mtDNA/ genomic DNA (ND1/Actin), (D-loop/Actin), or (ND1/18S), deletions containing the origin of replication/mtDNA (mito$Sca$I with $O_H$/D-loop), and deletions not containing the origin of replication/mtDNA (mito$Sca$I no $O_H$/D-loop). The following primers were used:

**mtDNA**. ND1 SYBR (F: CAGCCTGACCCATAGCCATA; R: ATTCTCCTTCTGT CAGGTCGAA)

D-loop SYBR (F: CCCCTTCCCCATTTGGTCTATT; R: TTGATGGCCCTGA AGTAAGAACC)

ND1 PrimeTime (1: GCCTGACCCATAGCCATAAT; 2: CGGCTGCGTATTC TACGTTA;

Probe: 56-FAM/TCTCAACCC/ZEN/TAGCAGAAACAAACCGG/3IABkFQ)

D-loop PrimeTime (1: TCTCGATGGTATCGGGTCTAA; 2: CTTGACGGCT ATGTTGATGAAA; Probe: 5TET/AGCCCATGA/ZEN/CCAACATAACTGTGG T/3IABkFQ)

mito$Sca$I with $O_H$ SYBR (F: CGCAAAACCCAATCACCTAA; R: GGGCTTG ATTTATGTGGT)

mito$Sca$I no $O_H$ SYBR (F: CAGCCCTCCTTCTAACAT; R: GTCATGAAATC TTCTGGGTGTAGG)

mito$Sca$I with $O_H$ and $O_L$ SYBR (F: ACTCACCAATATCCTCAC; R: GCCCCC TCAAATTCATTC)

**Genomic DNA**. β-Actin SYBR (F: GCGCAAGTACTCTGTGTGGA; R: CATCGT ACTCCTGCTTGCTG)

β-Actin PrimeTime (F: CTCCCTGGAGAAGAGCTATGA; R: CCAAGAAGG AAGGCTGGAAA; Probe: 5Cy5/TCATCACTATTGGCAACGAGCGGT/3IAbR QSp)

18 S PrimeTime (F: GCCGCTAGAGGTGAAATTCT; R: TCGGAACTACGAC GGTATCT;

Probe: 5Cy5/AAGACGGACCAGAGCGAAAGCAT/3IAbRQSp).

**Immunohisto- and immunocyto-chemistry**. Liver sections were cut at 12 µm thickness with a cryostat (Leica). For immunofluorescent staining, sections underwent antigen retrieval in 10 mM sodium citrate buffer. Sections were blocked for 1 hour in 5% BSA in PBS at RT. Sections were incubated with primary antibody (anti-HA 1:200 in 5% BSA (#ROAHAHA, Sigma)) for 16 hours at 4 °C. Sections were the incubated with secondary antibody for 2 hours at RT (Alexa-fluor anti-Rat/594 1:200 in 5% BSA (#A-11007, Molecular Probes)) and mounted with Vectashield hard set mounting medium with DAPI.

Live fibroblasts on coverslips were incubated with MitoTracker Red CMXRos (100 nM for 30 minutes) (Molecular Probes) then fixed with 4% PFA. Fibroblasts were permeabilized with ice-cold methanol and blocked for 1 hour in 5% BSA in PBS at RT. Coverslips were incubated with primary antibody (anti-HA 1:200 in 5% BSA (#ROAHAHA, Sigma)) for 16 hours at 4 °C. Coverslips were the incubated with secondary antibody for 2 hours at RT (Alexa-fluor anti-Rat/488 1:200 in 5% BSA (#A-11006, Molecular Probes)) and mounted with Vectashield hard set mounting medium with DAPI. Images were captured using a Zeiss LSM710 confocal microscope.

**Western blots**. Cell homogenates were prepared in PBS containing protease inhibitor mixture (Roche Diagnostics). Homogenates were snap frozen in liquid nitrogen, sonicated for 3 s, then centrifuged at 14,000×g at 4 °C, and the supernatant collected. Protein concentration was determined by Lowry assay using the BCA kit (BioRad). Approximately 20 µg of protein were separated by SDS-PAGE in 7.5% acrylamide gels and transferred to PVDF membranes. Membranes were blocked with 5% non-fat milk in 0.1% Tween-20 in PBS and subsequently incubated with anti-POLGA (#ab128899 Abcam). Antibody was used at 1/ 1000 dilution (in 0.5% milk in PBST). Secondary antibodies conjugated to horse-radish peroxidase (Cell Signaling Technologies) were used, and the reaction was developed by chemiluminescence using SuperSignal West reagent (Thermo Scientific).

**Sequencing mito$Sca$I rearrangement breakpoints**. To analyze the rearranged mtDNA molecules from [mito$Sca$I with $O_H$] and [mito$Sca$I with $O_H$ and $O_L$] PCR products from the mito$Sca$I $Polg$ mutator and control infected liver and fibroblasts were isolated and cloned using the TOPO TA 2.1 Cloning kit (Invitrogen). Clones were screened and positive clones were sequenced using the vector specific primers, M13F and M13R. DNA analyses were performed using CLC Bio (QIAGEN) software package.

**Data availability**. All the data described in this study is available upon request.

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

## Acknowledgements

This work was supported by the National Institutes of Health Grants 1R01AG036871, 5R01EY010804, and 1R01NS079965 (CTM). Nadee Nissanka is supported by an American Heart Association predoctoral fellowship (16PRE30480009) and Lois Pope LIFE Fellowship program. We acknowledge support from the NEI center grant P30-EY014801 from the National Institutes of Health (NIH).

## Author contributions

N.N. planned and performed most experiments and wrote manuscript. S.R.B. prepared adenovirus and helped inject in the retro-orbital sinus. M.J.P. helped perform lentivirus infection in fibroblasts. C.T.M. helped design experiments and wrote manuscript.

## Additional information

**Competing interests:** The authors declare no competing interests.

