## [Peer Review File · Nature Communications]

Reviewers' Comments:

Reviewer #1:

Remarks to the Author:

The article by Nissanka et al. asks the question how linearized fragments of mtDNA are eliminated in the mitochondria. They utilize as their tool the mtDNA Mutator mice, carrying an inactivated exonuclease domain of the mitochondrial DNA replicase, DNA pol gamma (POLG), leading to an accumulation of mtDNA point mutations and linearized mtDNA fragments, and to a progeroid syndrome. Linear mtDNA fragments, which should not be able to replicate, have been reported to be eliminated quickly. The proteins that are involved in the elimination have remained unknown, and even the existence of DSB-repair mechanisms in mitochondria have recently been questioned (Moretton et al. Plos One 2017).

In this report, the authors challenge the putative DSB repair/elimination system by increasing the formation of DSBs by adenoviral-mediated expression of mitochondrial targeted endonucleases. This innovative model of mito-targeted endonucleases to study mechanisms of mtDNA maintenance has been developed and utilized successfully during many years by the authors, i.e. Moraes group. The experimentation is of high quality and the results clearly presented. My only concern is that result-wise the paper is somewhat thin.

A recent paper by Moretton et al. 2017 (correctly referenced in the current manuscript) found no evidence for DSB repair mechanism in mitochondria. They also utilized mito-targeted endonucleases (a gift by Moraes group) in WT cells, and in their hands downregulation of known naturally occurring mitochondrial nucleases had no impact in elimination of linear mtDNA fragments, and they suggest that the elimination may occur by autophagic mechanisms. They did not, however, test POLG-Exo-activity.

The clear take-home message of the current report is that in fibroblasts from Mutator mice, as well as in vivo (in the liver of Mutator mice) linear mtDNA fragment removal requires polymerase gamma exo-activity. Retrospectively, this was also the obvious candidate protein, as the activity is lacking in the Mutators, and these mice are the only in vivo model where linear mtDNA fragments have been observed or reported. When the authors induced DSBs in Mutator fibroblasts, they found clear accumulation of mtDNA fragments only in the Mutators, but not in the wildtypes, emphasizing the efficient removal of linear fragments in the WTs. The finding was replicated with different mito-targeted endonucleases. Whether exo directly chops down the linear mtDNA fragments, or whether exo indirectly affects the cellular environment to promote DSB repair is not still clear. Does the processivity reported for POLG match with the rapid removal of the fragments?

Do the authors consider POLG-exo to work alone? If yes, the results suggest that POLG can load to DNA-template anywhere in mtDNA, even if the origins of replication are lacking. This is consistent with the Holt/Jacobs model of replication, which could be discussed.

Is POLG2 required for linear mtDNA elimination? ShRNA to take down POLG2 would be interesting.

The authors make a comment of relevance of their findings for mtDNA maintenance diseases. Exo-inactive mutations of POLG are not known in humans, whereas catalytic mutations (such as Y955C) are common. If the authors bring into their DSB-deficient cells a POLG with catalytic domain mutation and WT exo, can they rescue the defect of linear mtDNA elimination? The authors have a nice system that they could use a bit more in depth to characterize the DSB-repair mechanism in mitochondria.

Reviewer #2:

Remarks to the Author:

In this manuscript by Nissanka et al. the authors unravel a new mode of mtDNA degradation, carried out by the replicative polymerase POLG after double stranded breaks (DSBs). By the mean of adenoviral mediated delivery, the authors induce mtDNA DSBs using mitochondrial targeted enzymes: ScaI (blunt end, 3 cutting sites) and ApaLI (sticky end, 1 cutting site) in wild type mice and POLGD257A/D257A mice (mutator mice). The same experimental setup was used to induce mtDSBs in lung fibroblasts derived from animals with the same genotypes. In parallel the previously established mitoPstI was also tested, confirming the observations and pointing to a general mechanism.

The authors show that in wild-type settings, broken mtDNA molecules were efficiently degraded. Instead, they observe a significant delay in mtDNA degradation in the context of an exonuclease deficient POLG. The dynamics of the events were monitored by southern blot and quantitative real time PCR showing a good correlation between in vivo and ex vivo models, with a marked decrease in mtDNA copy number accompanied by the appearance of degradation products. Moreover, the authors detected mtDNA rearrangements in the form of deletions after breaks generated by mito ScaI. Consistently, the rearrangements were detectable only in the presence of an exonuclease deficient POLG. This underlines a possible mechanism by which these pathological mutations could foster mtDNA rearrangements and deletions observed in patients. The authors discuss the fate of these toxic byproducts.

The conclusions of this manuscript by Nissanka et al. are novel, of high interest, and likely to exert an immediate impact on the understanding of mtDNA instability disorders. The results presented are well supported by ex vivo and in vitro experiments, and suitable for the readership of Nature Communications.

1. In Figure 3A-B, the authors follow the dynamics of mtDNA degradation for both mtScaI and mtApaLI. If the nuclease degradation by POLG starts a DSB end, the prediction would be that the rate with which mtDNA disappears in setting of mtScaI (3 independent sites) cleavage would be much faster than those in which mtApaLI (1 cleavage site) is used. It would be important to show an absolute quantification of mtDNA content in mtScaI vs mtApaLI treated cells at earlier time points (ie 12, 18, 24, 36 hours). Following the rate of degradation at earlier time frames is informative.

2. The dynamics of degradation in the context of mutant POLG (Fig 2 and Fig 3) reveal a delay in mtDNA degradation. Interestingly the data does not show a complete block in mtDNA degradation. It would be important to test the involvement of other nucleases present in the mitochondria, including MGME1, MRE11, CtIP, FEN1, DNA2.

3. In Figure 4, the authors show very nicely the accumulation of mtDNA molecules bearing deletions, which were detected only in the presence of mutated POLG. It would be informative to address the role of LIG3 during the ligation step.

4. The two kinds of rearranged molecules (Figure 4) seem to be generated with very different efficiency ex vivo, but appear at the same frequency in vivo. Please Elaborate.

5. I suggest that the authors expand the sequencing results in Figure S3 using Nex-Gen sequencing approaches and to incorporate the results in the main figures.

6. In Suppl. Fig S3 rearrangement breakpoints are noted. It would be useful to know how many clones were sequenced and the relative frequency of different junctions. Additionally, in two out three cases, a single ScaI site is lost, potentially mimicking a recombination event between the palindromic repeats. Considering the importance of direct repeats in mediating deletions in patients with mtDNA disorders, this event should be explored and discussed.

Answer to Reviewers

Reviewer #1:

In this report, the authors challenge the putative DSB repair/elimination system by increasing the formation of DSBs by adenoviral-mediated expression of mitochondrial targeted endonucleases. This innovative model of mito-targeted endonucleases to study mechanisms of mtDNA maintenance has been developed and utilized successfully during many years by the authors, i.e. Moraes group. The experimentation is of high quality and the results clearly presented. My only concern is that result-wise the paper is somewhat thin.

We thank the Reviewer for the comments. We have added additional characterization of mtDNA deletions following DSB in Polg^{exo} cells. Although there are many follow-up questions, we focused this report on the effect of lacking Polg exonuclease activity has on generating mtDNA deletions. We believe this is a very important finding and have changed the title to reflect that.

*Whether *exo* directly chops down the linear mtDNA fragments, or whether *exo* indirectly affects the cellular environment to promote DSB repair is not still clear. Does the processivity reported for POLG match with the rapid removal of the fragments?*

We did not measure the speed of elimination in Polg wild-type background after DSB. The time points analyzed in these non-synchronized expression experiments (1-10 days after virus infection) would not allow us to make these determinations, as elimination occurs rapidly (less than 24 hours). We have previously reported that linear fragments were no longer visible after 2-4 hours mito*Apa*LI induction in cultured cells. Even at these time points the levels of linear fragments were extremely low.

*Do the authors consider POLG-*exo* to work alone? If yes, the results suggest that POLG can load to DNA-template anywhere in mtDNA, even if the origins of replication are lacking. This is consistent with the Holt/Jacobs model of replication, which could be discussed.*

Although we do not know if other factors are involved in the elimination of linear fragments, our results do suggest that Polg can load anywhere in the genome, even if the origins of replication are missing. We argue that in the discussion. We did amplify this discussion to address the Holt/Jacobs model as suggested by the Reviewer.

Is POLG2 required for linear mtDNA elimination? ShRNA to take down POLG2 would be interesting.

Although we agree with the Reviewer that this is an interesting question Polg2 is a factor required for mtDNA replication. Work from the Copeland lab showed that KO mice are embryonic lethal and early embryos had very low mtDNA levels. This is something we would like to investigate, but we feel that it is important to report this novel role for PolgA.

The authors make a comment of relevance of their findings for mtDNA maintenance diseases. Exo-inactive mutations of POLG are not known in humans, whereas catalytic mutations (such as Y955C) are common. If the authors bring into their DSB-deficient cells a POLG with catalytic domain mutation and WT exo, can they rescue the defect of linear mtDNA elimination? The authors have a nice system that they could use a bit more in depth to characterize the DSB-repair mechanism in mitochondria.

We agree with the Reviewer that this is a very interesting experiment, and in the “to do” list. However, we believe it should be part of a more comprehensive analysis in a future report.

Reviewer #2 (Remarks to the Author):

The conclusions of this manuscript by Nissanka et al. are novel, of high interest, and likely to exert an immediate impact on the understanding of mtDNA instability disorders. The results presented are well supported by ex vivo and in vitro experiments, and suitable for the readership of Nature Communications.

1. In Figure 3A-B, the authors follow the dynamics of mtDNA degradation for both mtScaI and mtApaLI. If the nuclease degradation by POLG starts a DSB end, the prediction would be that the rate with which mtDNA disappears in setting of mtScaI (3 independent sites) cleavage would be much faster than those in which mtApaLI (1 cleavage site) is used. It would be important to show an absolute quantification of mtDNA content in mtScaI vs mtApaLI treated cells at earlier time points (ie 12 , 18, 24, 36 hours). Following the rate of degradation at earlier time frames is informative.

We agree with the Reviewer that this will be interesting to find out. The times suggested by the Reviewer are difficult to work with because adenovirus expression is low before 24 hours. When considering 24 and 48 hours after infection, our quantifications do suggest that the ND1-containing ScaI fragment (smaller) underwent a more severe depletion than the corresponding ApaLI fragment (larger). We have added this comment to Results. In addition, we expanded the analyses of the breakpoint regions to determine whether there was increased “chewing back” in the breakpoints from WT cells compared to the MUT cells and found no differences.

2. The dynamics of degradation in the context of mutant POLG (Fig 2 and Fig 3) reveal a delay

in mtDNA degradation. Interestingly the data does not show a complete block in mtDNA degradation. It would be important to test the involvement of other nucleases present in the mitochondria, including MGME1, MRE11, CtIP, FEN1, DNA2.

We once again agree with the Reviewer. Some of these enzymes were tested in Moretton et al, 2015 (MGME1, FEN1, DNA2) as well as ENDOG and EXOG, and found not to degrade linear fragments. We are planning to follow the reviewer suggestion and explore this pathway in future studies.

3. In Figure 4, the authors show very nicely the accumulation of mtDNA molecules bearing deletions, which were detected only in the presence of mutated POLG. It would be informative to address the role of LIG3 during the ligation step.

Likewise, we believe this is an interesting follow-up question.

4. The two kinds of rearranged molecules (Figure 4) seem to be generated with very different efficiency ex vivo, but appear at the same frequency in vivo. Please Elaborate.

We have noted that as well, but do not have an explanation. In any case, we have performed additional experiments on another deletion that preserves both OH and OL. In this case, they accumulate at higher levels and increase with time, providing further evidence that the availability of free ends cause age-related and other pathogenic mtDNA deletions.

5. I suggest that the authors expand the sequencing results in Figure S3 using Nex-Gen sequencing approaches and to incorporate the results in the main figures.

We have followed the Reviewer's suggestion. We now include many more breakpoints in the analyses. A new figure (Fig.5) and expanded Supplementary figures were added.

6. In Suppl. Fig S3 rearrangement breakpoints are noted. It would be useful to know how many clones were sequenced and the relative frequency of different junctions. Additionally, in two out three cases, a single ScaI site is lost, potentially mimicking a recombination event between the palindromic repeats. Considering the importance of direct repeats in mediating deletions in patients with mtDNA disorders, this event should be explored and discussed.

We have expanded the Supplementary figure S3 and a new S4 were now added with the details requested by the Reviewer.

Reviewers' Comments:

Reviewer #1:

Remarks to the Author:

The authors have responded to my questions, revised the text, but did not directly address the key question whether POLG exo is actually the enzyme activity that is in charge of the elimination of linear mtDNA fragments. Introduction of the catalytically mutant, exo-proficient enzyme to the cells with DSB-induction would prove their claim and strengthen the paper. This is not a laborious experiment.

In Figure 2 there must be a problem in figure transfer, as the symbols above the figure ($\tau=j=\tau=\tau$ etc) are not understandable. If they are as meant, they should be explained in the legends.

Otherwise I'm happy with the revisions.

Reviewer #2:

Remarks to the Author:

In this revised manuscript by the Moraes group, the authors fail to address the minimal concerns and experiments suggested by reviewers.

As eluded to by reviewer #1, the paper is thin on results. While, I agree with the author that the findings are significant, a number of experiments needed to be done prior to publication. It is unfortunate that the authors failed to do that.

Following the dynamics of mtDNA degradation at earlier time point (point #1) and testing the involvement of Lig3 and other nucleases/helicases (points 2-3) should be addressed prior to publication.

Answer to Reviewers

Reviewer #1:

*Is POLG_{exo} actually the enzyme activity that is in charge of the elimination of linear mtDNA fragments. Introduction of the catalytically mutant, *exo*-proficient enzyme to the cells with DSB-induction would prove their claim and strengthen the paper. This is not a laborious experiment.*

We have performed the requested experiment. This was not a trivial experiment, as the mouse Polg is not as well characterized as the human. We consulted with experts in Polg who did not have the mouse gene cloned, but provided information on its predicted sequence. We cloned the cDNA for PolgA, which is 3,654 bp long, using synthetic DNA fragments. We created wild-type and a polymerase mutant version (R920H). This mutation has been described previously in patients. The catalytic activity of this variant was determined experimentally, and this mutation was found to be essentially a polymerase null (Graziewicz et al., 2004). We independently cloned lentiviral vectors expressing the two different forms of PolgA and confirmed the synthesis of full length proteins using a commercially available anti-PolgA antibody. Following infection with the different lentiviral Polg variants, we created DSBs in the mtDNA using Ad-mitoScaI-HA and followed the mtDNA levels over time. We found no differences between the two POLG variants with respect to degradation dynamics (Supplemental Figure S3). This suggested that the polymerase activity does not play a role in degrading linear mtDNA fragments.

In Figure 2 there must be a problem in figure transfer, as the symbols above the figure (=t=j=t=t etc) are not understandable. If they are as meant, they should be explained in the legends.

We are grateful to the reviewer for pointing this out to us. This was an issue with the figure transfer after uploading into the system. We believe we solved the issue.

Reviewer #2:

Following the dynamics of mtDNA degradation at earlier time points.

To better address mtDNA degradation dynamics at early time points, we now expressed Ad-mitoScaI-HA in wild-type and mutator fibroblasts and collected cells at 1, 2, 3, 4, 5, 6, 8, 10, and 12 hours after infection. Using TaqMan technology, we studied mtDNA degradation dynamics comparing the degradation of two different fragments following mitoScaI expression (smallest fragment containing D-loop and largest fragment containing ND1). We see in both wild-type and mutator fibroblasts, the degradation of the smallest fragment occurs faster than that of the largest fragment (Figure 3e).

Testing the involvement of Lig3 and other nucleases/helicases should be addressed prior to publication.

Although we agree that testing the involvement of Ligase 3 as well as other nucleases and helicases in degrading linear mtDNA fragments following DSBs would be interesting and important, we would not be able to do these experiments in a relatively short period of time. In contrast to human, antibodies against mouse proteins are not as good, making it difficult to confirm the knockdown of these proteins. Still, we have tried to knockdown LIG3, MGME1, and TWNK using lenti-shRNA and siRNA and have only seen ~40% knockdown via qRT-PCR. In addition, during this review period a paper published in *Nat. Commun.* (Matic et al., 2018) showed that MGME1 has a role in degrading linear mtDNA fragments. We added this information to the discussion.

References:

Graziewicz MA, Longley MJ, Bienstock RJ, Zeviani M, Copeland WC. Structure-function defects of human mitochondrial DNA polymerase in autosomal dominant progressive external ophthalmoplegia. *Nat Struct Mol Biol* 11, 770-776 (2004).

Matic S, et al. Mice lacking the mitochondrial exonuclease MGME1 accumulate mtDNA deletions without developing progeria. *Nat Commun* 9, 1202 (2018).

Reviewers' Comments:

Reviewer #1:

Remarks to the Author:

The authors have now responded satisfactorily to all my major comments, and I have no further concerns.

Reviewer #2:

Remarks to the Author:

I am in favor of publication. the authors addressed my concerns.